# Analysis of AlphaFold and molecular dynamics structure predictions of mutations in serpins

**Pedro Garrido-Rodríguez** [1☺], **Miguel Carmena-Bargueño** [2☺], **María Eugenia de la Morena-Barrio** [1], **Carlos Bravo-Pérez** [1], **Belén de la Morena-Barrio** [1], **Rosa Cifuentes-Riquelme** [1], **María Luisa Lozano** [1], **Horacio Pérez-Sánchez** [2]*, **Javier Corral** [1]*

1 Servicio de Hematología y Oncología Médica, Hospital Universitario Morales Meseguer, Centro Regional de Hemodonación, Universidad de Murcia, IMIB-Arrixaca, CIBERER-ISCIII, Murcia, Spain, 2 Structural Bioinformatics & High Performance Computing Research Group (BIO-HPC), Universidad Católica de Murcia (UCAM), Murcia, Spain

☺ These authors contributed equally to this work.
* hperez@ucam.edu (HPS); javier.corral@carm.es (JC)

**Data Availability Statement:** Code used to run AlphaFold is freely available on DeepMind's GitHub (see references). Software used for MD is property

## Abstract

Serine protease inhibitors (serpins) include thousands of structurally conserved proteins playing key roles in many organisms. Mutations affecting serpins may disturb their conformation, leading to inactive forms. Unfortunately, conformational consequences of serpin mutations are difficult to predict. In this study, we integrate experimental data of patients with mutations affecting one serpin with the predictions obtained by AlphaFold and molecular dynamics. Five *SERPINC1* mutations causing antithrombin deficiency, the strongest congenital thrombophilia were selected from a cohort of 350 unrelated patients based on functional, biochemical, and crystallographic evidence supporting a folding defect. AlphaFold gave an accurate prediction for the wild-type structure. However, it also produced native structures for all variants, regardless of complexity or conformational consequences *in vivo*. Similarly, molecular dynamics of up to 1000 ns at temperatures causing conformational transitions did not show significant changes in the native structure of wild-type and variants. In conclusion, AlphaFold and molecular dynamics force predictions into the native conformation at conditions with experimental evidence supporting a conformational change to other structures. It is necessary to improve predictive strategies for serpins that consider the conformational sensitivity of these molecules.

## Introduction

Serpins are a superfamily of proteins sharing a common, strongly conserved structural configuration required to inhibit serin proteases. Even serpins with no inhibitory activity share a single common core domain consisting of 3 β strands, 7–9 α helices, and a reactive center loop (RCL) for the interaction with target proteases [1]. Serpins are folded into a native conformation with 5 β strands in the central A sheet and a RCL, which means a metastable stressed structure with inhibitory activity (Fig 1A). This configuration allows serpins to change their structure to a relaxed hyperstable form with 6 β strands in response to stimulus, usually by the

of Schrödinger LCC. Thus, access to it might be restricted. GROMACS and matplotlib codes are available at https://manual.gromacs.org/current/download.html and https://matplotlib.org/stable/users/installing/index.html, respectively. Variants p. Arg79Cys, p.Pro112Ser, p.Met283Val, p. Pro352insValPheLeuPro and p. Glu241_Leu242delinsValLeuValLeuValAsn ThrArgThrSer, mentioned in our work, are deposited in UniProt as VAR_007037 , VAR_086227, VAR_027468, VAR_086198 and VAR_086197, respectively.

**Funding:** JCC received funding in the form of a grant from Instituto de Salud Carlos III (PI21/00174; PMP21/00052; JR22/00041) and from Fundación Séneca (21886/PI/22). CBP received funding in the form of a grant from Instituto de Salud Carlos III (JR22/00041). The funders had no role in study design, data collection and analysis, decision to publish, or preparation of the manuscript.

**Competing interests:** The authors have declared that no competing interests exist.

cleavage of the RCL by their target protease (Fig 1C) [2]. This efficient suicidal mechanism explains the key role of serpins in several crucial systems based on proteolytic cascades in a wide range of species. It is the case of coagulation, inflammation or the complement system. Such structural sensitivity also makes serpins particularly vulnerable to even minor modifications caused by environmental conditions or missense mutations, which may cause a conformational instability with pathogenic consequences [3], as other hyperstable relaxed conformations with no inhibitory activity may be generated, such as latent structure (when the RCL is inserted into the own molecule) or polymers (if the insertion involves another molecule), both without inhibitory activity (Fig 1D). Moreover, polymers may be toxic if accumulated in the cell [4]. These aberrant conformations, caused by a wide range of mutations affecting *SERPINA1*, *SERPINC1*, *SERPINI1* or *SERPING1*, are responsible for disorders of relevance, such as emphysema, thrombosis, neuropathies or angioedema, respectively [5]. Moreover, environmental conditions may also induce conformational changes in serpins [6].

One of the best-known serpins is antithrombin, probably the most important endogenous anticoagulant [7]. Its deficiency dramatically increases the risk of thrombosis, even when it impacts a single allele, acting as a dominant disorder. Antithrombin deficiency can be classified into two groups. Type I when the mutation, by different mechanisms, severely affects the levels of variant antithrombin in plasma. Type II, if the mutation does not severely impair the secretion of the antithrombin variant which has impaired or null anticoagulant activity. More than 386 different variants affecting *SERPINC1*, the gene encoding antithrombin, have been described as causative of antithrombin deficiency [8].

One great challenge for antithrombin and all other serpins is how to predict the consequences of any new mutation, as they may have a wide range of consequences with clinical impact. Thus, certain mutations may cause intracellular polymerization that cause severe type I deficiency with serious clinical impact, others favor the transition to latent conformation with no functional activity, being responsible for type II PE deficiency also with severe clinical phenotype, and other mutations only disturb functional domains of the serpin (heparin binding domain or the RCL), leading to type II deficiency with usually milder clinical impact [8], being particularly interesting to identify those with a conformational effect [9].

Overall, the prediction of 3D protein structures based on their sequences is a long-lasting problem in biochemistry [10]. Likewise, it has also been a noteworthy challenge for artificial intelligence (AI) systems since the advent of bioinformatics, and we can see nowadays that AI methods have started to slightly increase the accuracy of structural predictions [11]. In 2021, Jumper *et al.* [12] released AlphaFold, their AI model for protein structure prediction. The model developed by DeepMind has created high expectations in the community [13] because of its outstanding results on the 13th edition of Critical Assessment of Structure Prediction (CASP) [14] using the first version of the model [15], and more recently in CASP 14 [16] with the new AlphaFold 2 [12]. The model, now released to the community, is expected to ease the efforts needed to solve the 3D structures of several proteins yet unknown [17]. Nonetheless, the structural resolution of non-novel, mutated proteins could mean a huge step forward in several areas. Although its authors state it is not validated to predict mutational effects [18] and flipping an amino acid in the target protein to model a mutation will not work in AlphaFold [19, 20], it would be interesting to challenge the model to solve the conformational consequences of mutations affecting serpins, as these mutations cause a relevant structural change compared with the native conformation.

Molecular Dynamics (MD) simulation is an established and routinary methodology for the prediction of the dynamical evolution of biomolecular structure with atomic detail [21]. Indeed, the first MD simulation of a simple protein system was carried out in 1977 [22], while recently MD allowed the simulation of the SARS-CoV-2 spike protein, implying millions of

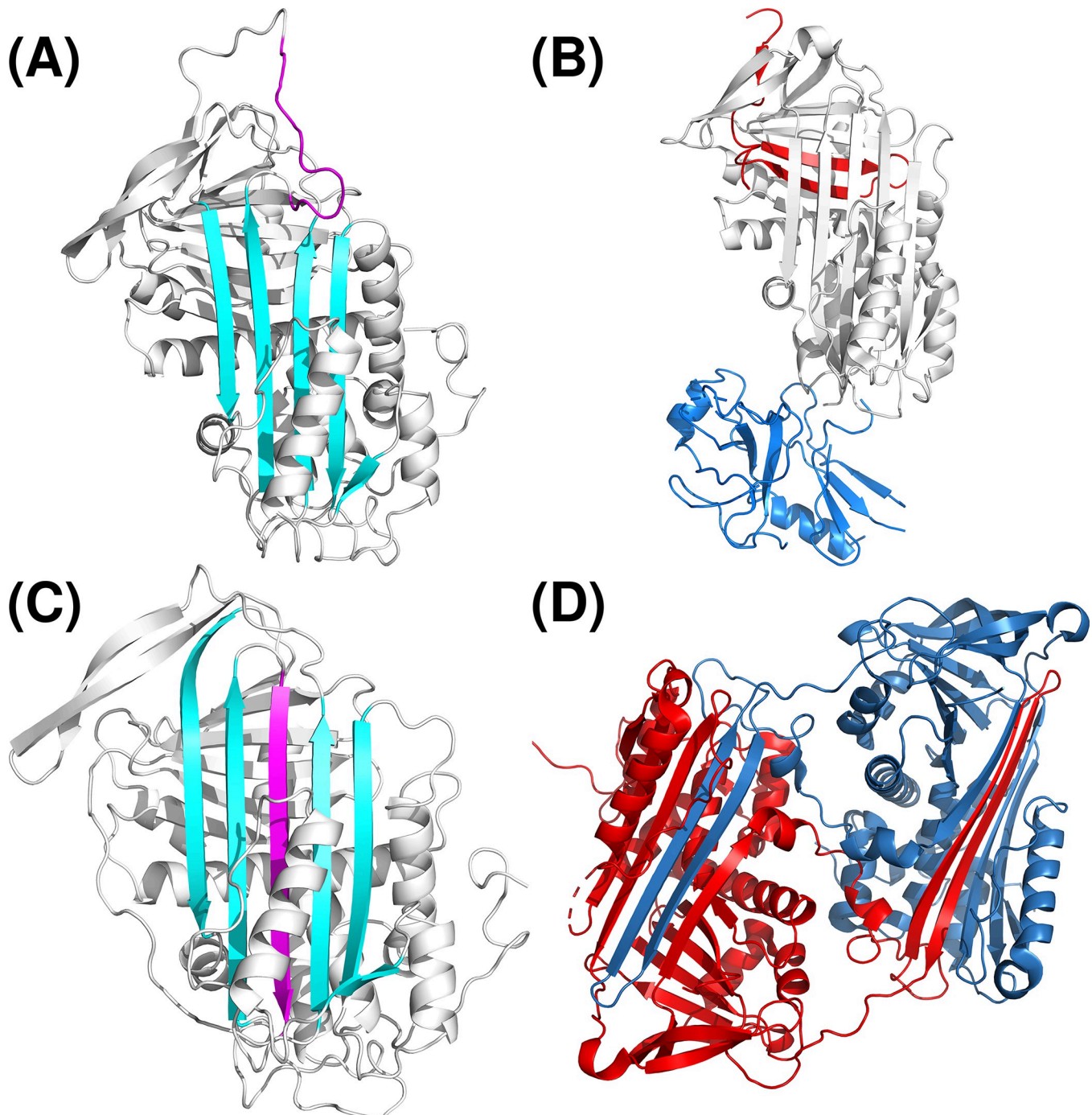

**Fig 1. Serpins conformations.** (A) Native or stressed conformation (PDB: 1AZX-I). (B) Cleaved state (PDB: 1EZX). (C) Latent or relaxed conformation (PDB: 1AZX-L). (D) Dimer structure, exchanging two central β-strands (PDB: 2ZNH). White, red and dark blue: serpin (antithrombin or α-1-antitrypsin). Cyan: central wild-type β-strands. Magenta: reactive center loop. Light blue: trypsin.

atoms [23]. Nonetheless, it should be mentioned that the main limitations of the MD technique are related, among others, to the required computing time, which depends mainly on the system size and the trajectory length needed. Advances in hardware in the last decade allow nowadays to perform millisecond simulations for proteins of average size such as p53

[24]. While parallelization techniques have significantly advanced the field of MD simulations, allowing for more complex and longer simulations, there remain inherent challenger in modelling time-dependent processes even with parallelization [25].

Thus, in this study we have explored the accuracy of AlphaFold and molecular dynamics predictions for different environmental conditions inducing conformational changes in antithrombin, as well as for natural mutants of antithrombin identified in our cohort of patients. All selected cases have variant antithrombins detected in plasma, ensuring both folding and secretion of variants, and were deeply studied using different biochemical methods providing information on their conformation, including a case whose crystal structure was determined.

## Materials and methods

### Patients

For 23 years (1998–2021) our group has recruited one of the largest cohorts of unrelated patients with antithrombin deficiency (N = 350). After molecular analysis of *SERPINC1*, the gene encoding this key anticoagulant, using sequencing methods and Multiple Ligand Probe Amplification (MLPA) methods to identify small and gross genetic variants, 135 different *SERPINC1* gene defects were found in 250 patients. Experimental characterization of plasma antithrombin in these cases, described extensively elsewhere [26–28] included functional analysis of anti-FXa and anti-FIIa activities by using chromogenic methods. These functional assays evaluated the inhibitory activity of plasma antithrombin. Moreover, in all these samples we also quantified antithrombin antigen levels by immunological methods, identified forms with low heparin affinity by crossed immunoelectrophoresis in the presence of heparin, and evaluated plasma antithrombins by native (in the presence and absence of 6M urea) and denaturing PAGE and western blotting, procedures that allow detection of aberrant forms of antithrombin and a semi-quantitative determination of the latent conformation [29].

For this study, we selected 5 different variants (Table 1). The rational of this selection was: 1) the variant generated antithrombin variants detected in the plasma of carriers; and 2) availability of biochemical experimental data, which may include the recombinant expression in eukaryotic cells of the mutated variant [30], its purification from plasma and in one case a crystallographic characterization. Moreover, this selection also aimed to cover the range of different mutations, from simple missense to the insertion of a different number of residues in the structure of antithrombin (Fig 2).

**Table 1. Functional antigenic and biochemical data of *SERPINC1* mutations identified in patients with antithrombin deficiency and selected for predictions using AlphaFold.**

| Mutation | ID | N | Anti-FXa (%) | Antigen (%) | Low Hep Aff CIE | Latent Native Urea | Dimers | Ref | Other data |
|---|---|---|---|---|---|---|---|---|---|
| p.Arg79Cys | M1 | 19 | 42.6 ± 9.8 | 91.6 ± 11.1 | Yes | -- | -- | [31] | -- |
| p.Pro112Ser | M2 | 1 | 50 | 55 | -- | -- | Yes | [32] | -- |
| p.Met283Val | M3 | 1 | 76 | 98 | Yes | Yes | – | [26] | -- |
| p.Pro352insValPheLeuPro | M4 | 5 | 47.5 ± 8.9 | 58.0 ± 1.7 | -- | -- | Yes | [33] | Verified by proteomic |
| p. Glu241_Leu242delinsValLeuValLeuValAsnThrArgThrSer | M5 | 1 | 57 | 80 | Yes | Yes | – | [34] | Crystal structure |

ID: variant alias; N: number of carriers; Anti-FXa: anti Factor Xa activity (% compared to normal levels); Antigen (% compared to normal levels); Low Hep Aff: low heparin affinity forms observed by crossed immunoelectrophoresis; Ref: article describing the variant.

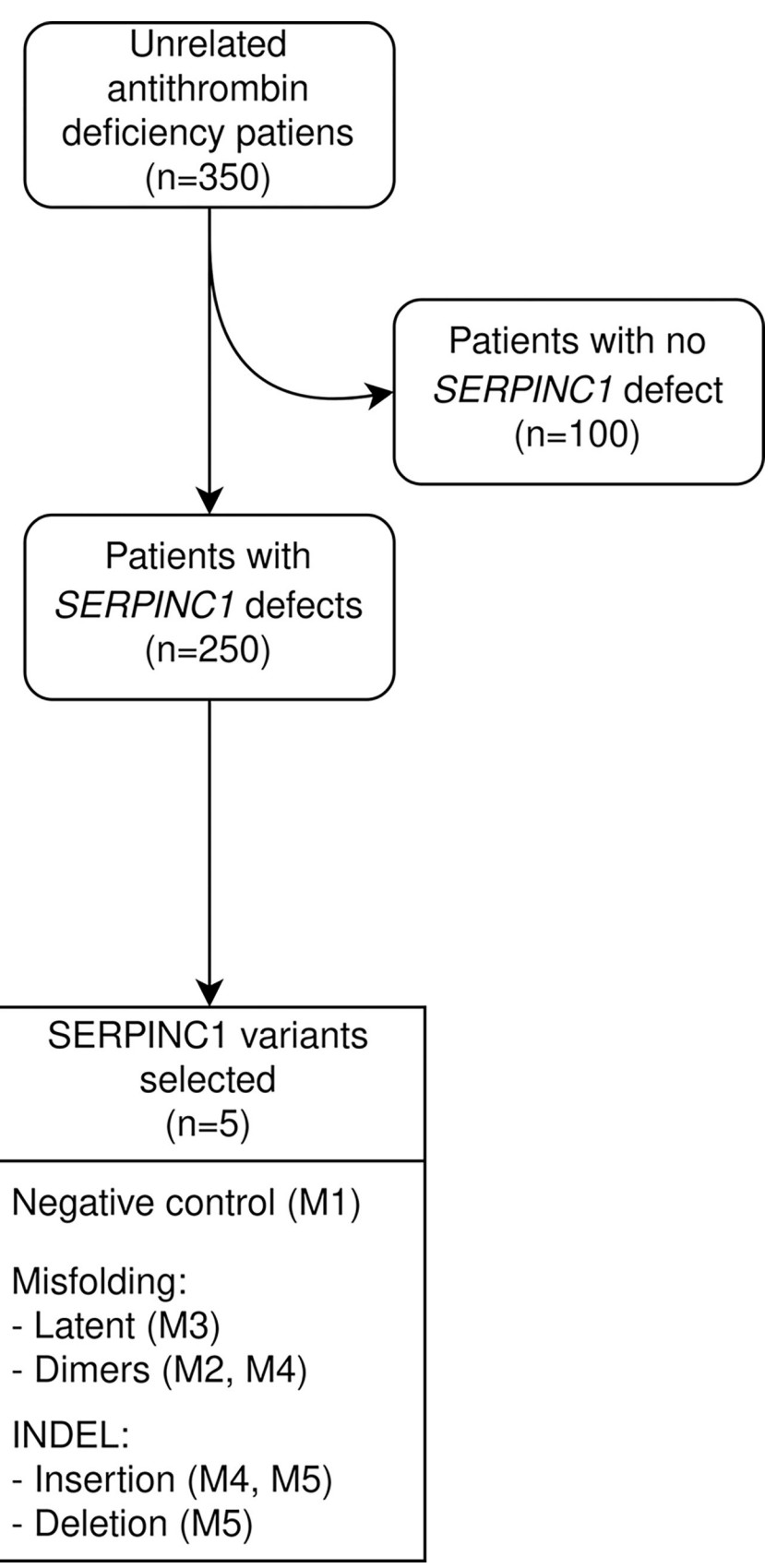

**Fig 2. Selection of variants for this study.** Variants were selected from a cohort of 350 patients with antithrombin deficiency and were previously described in different works.

## Structure processing

The sequences in FASTA of all proteins (wild-type and M1-M5) were used to obtain Alpha-Fold predictions. The 3D crystal structures were obtained from the Protein Data Bank (PDB) with codes 1AZX (wild-type) and 4EB1 (M5). These structures were preprocessed with Maestro Tool Protein Preparation Wizard [35] to fix potential problems with hydrogens that could crash between them. Proteins were processed to add charges using Maestro System Builder tool with OPLS3e force field.

## AlphaFold predictions

Monomer predictions were obtained running AlphaFold v2.0.1 [12, 36], on its simplified version, i.e., with no templates and a reduced Big Fantastic Database (BFD). Dimerization simulations were obtained with AlphaFold Multimer model [37]. UCSF ChimeraX v1.3 [38] was used to obtain metrics for AlphaFold results. Pruned RMSD refers to RMSD computed excluding from the process unstructured or flexible structures to avoid potential biases on the RMSD computation. These results were represented using PyMol v2.5. Hydrogen bond analysis was performed with ChimeraX as described elsewhere [38, 39].

We decided to run the aforementioned configuration for AlphaFold as (i) it requires less computational resources; (ii) AlphaFold authors claim accuracies to be nearly identical to the regular v2.0.1 for most of the proteins [36], we see no differences between both configurations, i.e. reduced and full AlphaFold (Table 2 and Fig 3); and (iii) antithrombin belongs to the serpin protein superfamily. That means it is not part of the reduced group of proteins showing significant differences between the complete and simplified model, as serpin proteins all have a very similar, conserved structure and are well represented in the databases used to build BFD. Aiming to evaluate the effect of each mutation, we run a script of Rosetta Online Server that Includes Everyone (ROSIE 2) [40] to stabilize proteins with a point mutation (https://r2.graylab.jhu.edu/apps/submit/stabilize-pm) and we compared the results obtained by both methods. The difference of free energy ($\Delta\Delta G$) of each protein was calculated in Rosetta Energy Unit (REU). A value of $\Delta\Delta G$ higher than 0 indicates that's exists a destabilizing mutation.

## Molecular dynamics

All considered protein structures were subjected to MD simulations with MD engine Desmond [41] included in the Maestro Suite [42]. First, all protein models were immersed in a box, whose dimensions were 10x10x10 Å, with water molecules with the simple point charge (SPC) water model. Ions of Sodium and Chlorine were added to neutralize charges and to obtain a molarity of 0.15 M. This step made use of Maestro System Builder tool. Complexes were passed to MD with the following parameters: energy minimization was made by 2000 steps using the steepest descent method with a threshold of 1.0 kcal/mol/Å. The NPT

**Table 2. Summary of AlphaFold metrics between predictions of reduced and full (reference) versions.** We observe no differences between full and reduced AlphaFold models.

| Protein | RMSD mean (Å) | RMSD maximum (Å) | lDDT |
|---------|---------------|------------------|------|
| Wild-type | 0.015 | 0.030 | 0.995 |
| M5 | 0.014 | 0.030 | 0.999 |

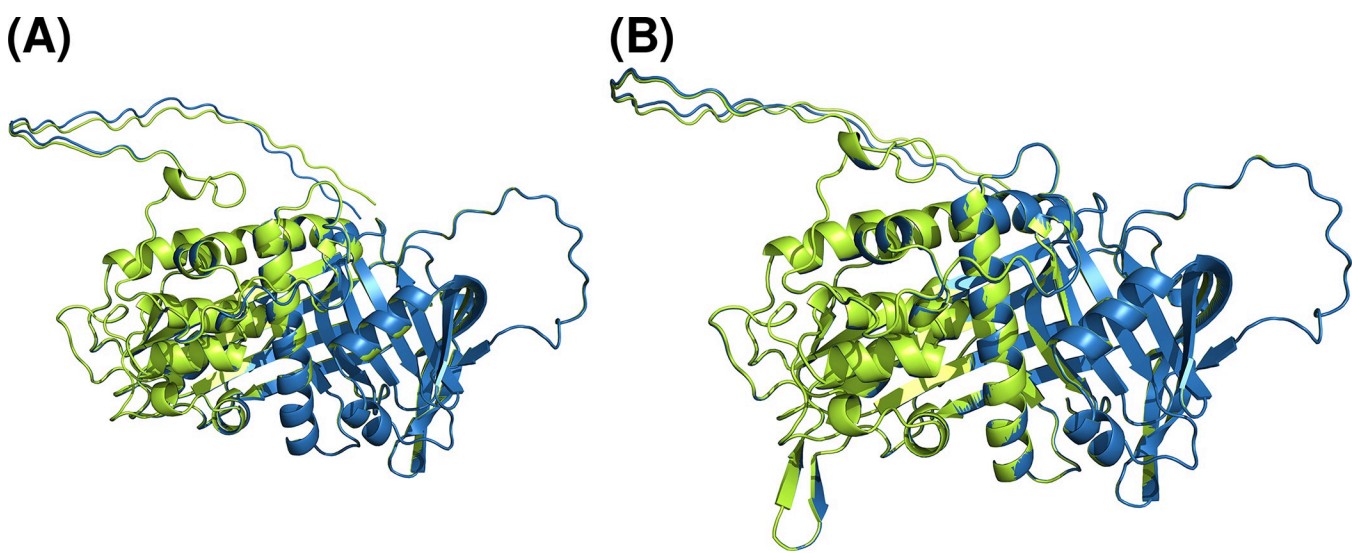

**Fig 3. Results for reduced (green) and full (blue) versions of AlphaFold.** (A) Wild-type antithrombin predictions. (B) M5 predictions. No major differences are perceived for antithrombin between reduced and full AlphaFold versions.

simulations were realized at the exact temperature in each case: 300, 313 and 373 K with the Nosé-Hoover algorithm [43, 44] and the pressure was maintained at 1 bar with the Martyna-Tobias-Klein barostat [44, 45]. The length of performed MD simulations was 100ns or 1000ns. Periodic conditions were used. The cutoff of 9 Å was established to van der Waals interactions and the Particle Mesh Ewald (PME) method with a tolerance of 10–9 was used in the electrostatic part. The force field used in all runs was OPLS3e [46].

## Metrics

In order to evaluate the accuracy of the structural predictions obtained by AlphaFold and MD simulations, the Secondary Structure Elements (SSE), the Root Mean Squared Deviation (RMSD), the Root Mean Squared Fluctuation (RMSF) and the Local Distance Difference Test (lDDT) [47] metrics were considered for each protein model, when comparing against available crystal structure. To evaluate the changes produced in the secondary structure of proteins along the trajectory, Maestro Simulation Interaction Diagram (SID) tool was used under version 2020–04, in order to show the evolution of α-helices and β-strands along the whole simulation. This tool analyzes many properties of the protein, such as RMSD, RMSF and SSE. SID shows a summary of SSE in all residues of the protein, so this analysis can be used to evaluate if a residue (or a group of residues) changes its secondary structure along the simulation. RMSD evolution for each protein was computed using SID. RMSD is a measurement of the global stability of the protein along the simulation. RMSF evolution for each protein was also computed using SID. RMSF measures the stability of each residue along the whole simulation. To compute both RMSD and RMSF, SID takes the first frame as reference and superimposes all the frames against the aforementioned to evaluate protein fluctuations along the trajectory. In each case, values over 3 Å mean that the protein (RMSD) of a specific region (RMSF) is not stable. lDDT was performed using lDDT Swiss-Model Tool (https://swissmodel.expasy.org/lddt/downloads). This score is a measurement of the difference between the local interactions of two proteins. This tool was used in the CASP9 (Critical Assessment of protein Structure Prediction 9: https://predictioncenter.org/casp9/index.gci).

In addition, Free Energy Landscape (FEL) analysis combined with PCA and the covariance matrix generation were calculated using GROMACS 2022.2 [48] modules named *covar* and *anaeig* when processing Desmond MD trajectories via Maestro Tool: trj convert [11]. Subsequently, 3D FEL figures were obtained using the Python library *matplotlib* 3.5.2 [49]. Pathogenicity predictions were gathered for each variant when available using Varsome v11.3 [50]. Additionally, Ramachandran plots were generated for each structure analyzed as a quality control step (S1 Fig). The dihedral angles and G-factor were calculated using BioLuminate software included in Maestro Suite [51].

## Ethical considerations

All methods and experimental protocols used in this study have been carried out following current guidelines and regulations. This study was approved by the local Ethics Committee of Morales Meseguer University Hospital and performed in accordance with the 1964 Declaration of Helsinki and its later amendments. All included subjects and/or their legal guardian(s) gave their written informed consent to enter the study. Patients' data were gathered between 24/01/2022 and 27/01/2022. Researchers did not have access to information that could identify individual participants during or after data collection.

## Results

### Mutations evaluated

Table 1 shows the mutations selected, the consequences on the protein and the experimental data obtained in each variant.

We selected three missense mutations:

1. p.Arg79Cys (M1). This mutation causes a type II HBS deficiency (Antithrombin Toyama variant) [31], as it changes one arginine residue directly involved in the interaction with heparin [52], the cofactor of antithrombin that fully activates this serpin increasing up to 1000-fold its anticoagulant activity [53]. Moreover, this is one of the most recurrent variants identified in our cohort (N = 19 unrelated cases). All carriers of this variant have reduced heparin cofactor activity, normal antigen levels and the mutated antithrombin (which constitutes half of the plasma antithrombin) has faster electrophoretic mobility in native-PAGE and low heparin binding under crossed immunoelectrophoresis with heparin. Finally, this residue has also been mutated to Ser or His in other patients with type II HBS deficiency.

2. p.Pro112Ser (M2). This mutation causes a severe reduction of antithrombin levels in plasma, probably by inducing intracellular polymerization according to the detection of disulfide-linked dimers in plasma, as our group previously demonstrated [32].

3. P.Met283Val (M3). The patients carrying this mutation had high antigen levels, with the increase of a form with low heparin affinity, and increased levels of latent antithrombin as demonstrated by our group, all data supporting a type II PE deficiency [26].

We also included in our study two mutations with greater consequences in the amino acid sequence of antithrombin:

1. C.1154-14G>A, which created a new splicing acceptor site in intron 5. The resulting mRNA maintained the reading frame of antithrombin with the insertion of 4 new residues p.Pro352insValPheLeuPro (M4) [33]. This new variant might induce polymerization of the variant antithrombin, as a severe reduction of antigen levels was observed in 5 unrelated carriers of this recurrent mutation. We also detected the presence of disulfide-linked dimers

in plasma, which were slightly bigger than those identified in carriers of the p.Pro112Ser mutation. The purification and proteomic analysis of disulfide-linked dimers from carriers' plasma confirmed the insertion of the predicted 4 additional amino acids in the variant antithrombin [54].

2. c.722_725delins[731_751;GAACCAG], which caused a complex insertion p.Glu241_-Leu242delinsValLeuValLeuValAsnThrArgThrSer (M5) in a very conserved region of serpins. The carrier of this mutation had a type II deficiency with increased levels of a form with low heparin affinity and hyperstable antithrombin. Moreover, structural analysis with the recombinant protein revealed a relaxed structure with the inserted residues forming a new strand in the central A sheet and a native RCL [34].

S1 Table shows pathogenicity prediction for selected mutations.

## AlphaFold predictions

Firstly, we compared the structure of native antithrombin (PDB:1AZX) with the one predicted by AlphaFold for the wild-type sequence of human antithrombin (https://www.uniprot.org/uniparc/UPI000002C0C1). As shown in Fig 4A, AlphaFold predictions were quite accurate, despite some differences concerning the starting or end of helix or strands. Differences are summarized as RMSD in Table 3.

Then, we compared AlphaFold prediction for the wild-type antithrombin with the predictions of the same software for all selected mutants. As expected, the functional variant M1 only caused a minor modification of the structure, even at the mutated residue (Fig 4B and Table 3).

The comparison of the other two missense mutations included in this study was quite similar, with minor mismatches (Fig 4C, 4D and Table 3). These three mutations (M1, M2 and M3) were analyzed using Rosetta Online Server that Includes Everyone (ROSIE 2). The results showed that all mutations have a value of REU higher than the native (Table 4), thus all were destabilizing, especially M2, with an energy difference of 5.88 REU. M1 and M3 have an energy difference close to 2 REU. Furthermore, we compared ROSIE 2 predictions with AlphaFold's. These comparisons are summarized in Table 4.

Interestingly, AlphaFold also produced a native structure for variants causing insertions of residues. For the small insertion of 4 amino acids caused by the intronic mutation (M4), AlphaFold predicted a small elongation of the loop connecting hI and s5A (Fig 4E and Table 3).

For the M5 variant, the inserted new residues are forced into a new small helix but maintained the native structure (Fig 4D), which differed significantly from the crystal structure of this variant (4EB1) (Fig 5 and Table 3). The analysis of dihedral angles revealed that residues Ala206, Ile207 and Asn208 adopted different secondary structures in the M5 crystal compated to the AlphaFold prediction. In M5 crustal these residues are part of a β-strand, whereas in the AlphaFold they form a right-handed α-helix (Tables 5 and 6). The information of the Chi1, Chi2 and G-factor are summarized in S2 and S3 Tables.

When analyzing changes in the hydrogen bond network (HBN) of the different structures, we observed little differences between the hydrogen network of AlphaFold's wild-type and its predictions for point mutations (Table 7), sharing nearly 90% of intra-chain connections via hydrogen bonds. Nonetheless, when comparing 1AZX HBN with 4EB1 and M5, we find the latter having more in common with the native crystal than 4EB1. The differences between the HBN of M1-M3 and M4-M5 arise from the insertion of new residues, creating and destroying hydrogen bonds, thus changing the HBN.

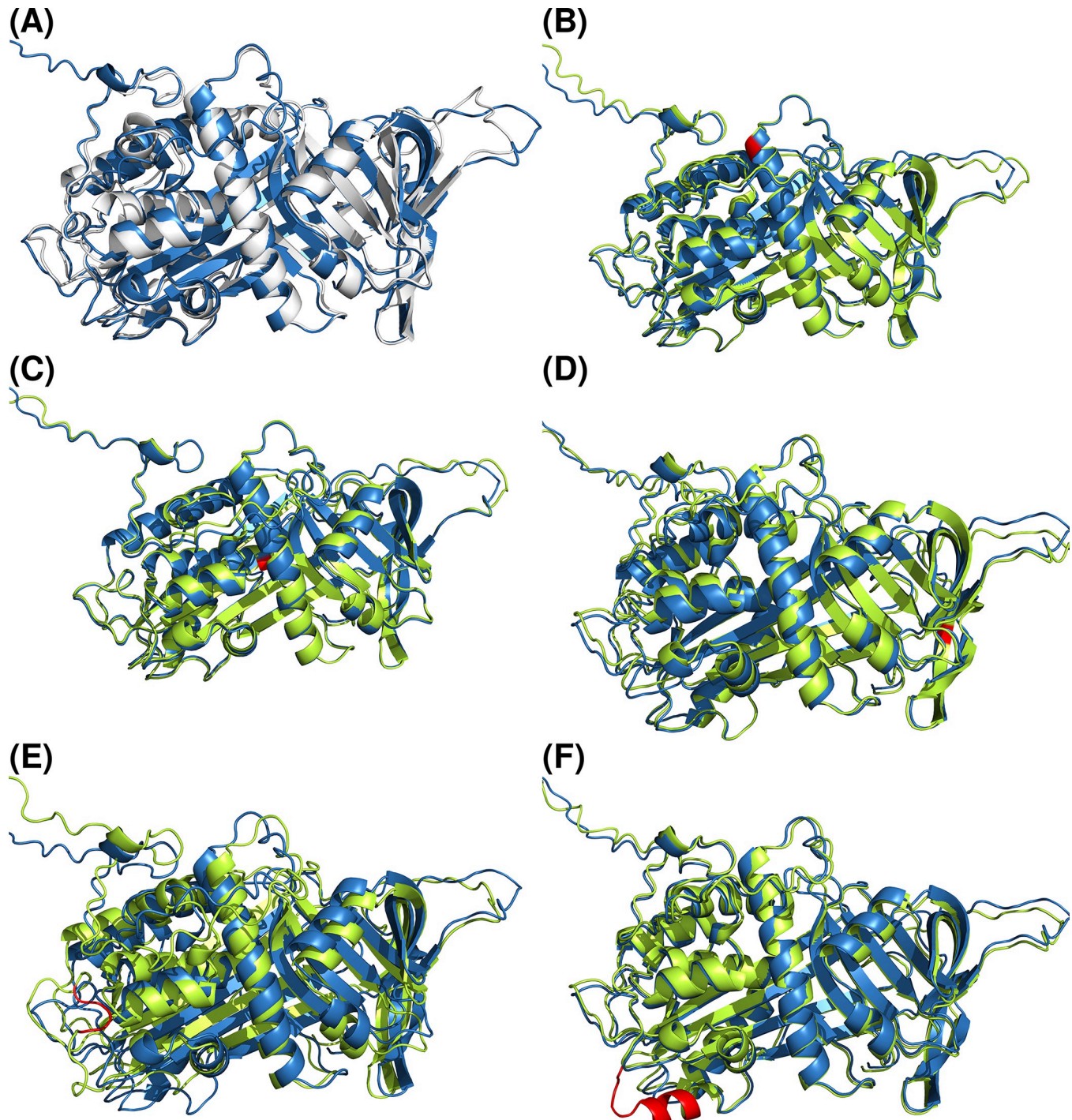

**Fig 4. Comparisons between wild-type antithrombin prediction and different structures.** (A) 1AZX. (B) M1. (C) M2. (D) M3. (E) M4. (F) M5. White: 1AZX. Blue: AlphaFold prediction for wild-type antithrombin. Green: corresponding mutation. Red: mutated residues. Strong correlation between wild-type, native structure and mutant predictions is observed, even for bigger changes, such as M4 and M5. M4 is included as a loop in the main structure, whilst M5 is modeled as an α-helix, as this configuration has higher thermodynamical stability than a 10-residue loop.

**Table 3. Summary of metrics for AlphaFold and molecular dynamics.** AlphaFold metrics are referred to its predictions, except for PDB:1AZX and PDB:4EB1. Molecular Dynamics metrics refer to differences between initial and final state of the proteins.

| AlphaFold | | | |
|---|---|---|---|
| **Structure A** | **Structure B** | **RMSD (Å) pruned** | **RMSD (Å) all pairs** |
| Wild-type | PDB:1AZX | 0.858 | 2.431 |
| | M1 | 0.302 | 2.628 |
| | M2 | 0.320 | 2.999 |
| | M3 | 0.377 | 2.812 |
| | M4 | 0.504 | 7.355 |
| | M5 | 0.347 | 1.675 |
| PDB:4EB1 | | 0.940 | 4.003 |
| | Wild-type | 0.933 | 3.890 |
| Molecular Dynamics | | | | | |
| **Simulation time (ns)** | **Protein** | **RMSD (Å)** | **lDDT** | **RMSF man (Å)** | **RMSF maximum (Å)** |
| 1000 | Wild-type | 1.785 | 0.716 | 1.005 | 6.013 |
| | M3 | 1.685 | 0.733 | 1.008 | 6.546 |
| 100 | M1 | 1.422 | 0.725 | 1.005 | 3.860 |
| | M2 | 1.748 | 0.729 | 1.104 | 5.564 |
| | M4 | 1.481 | 0.733 | 1.012 | 4.035 |
| | M5 | 1.668 | 0.710 | 1.022 | 5.091 |

Also, we evaluated the differences between the reduced and complete version of AlphaFold with both wild-type protein and M5. We selected the wild-type sequence for benchmarking as AlphaFold has been validated with wild-type proteins, and M5 as it is the most aberrant and challenging variant for modeling on our study. Resulting metrics (Table 2) showed virtually nonexistent differences between both versions of AlphaFold, validating thus the applicability of the reduced version of AlphaFold, as its results are comparable to those obtained with the complete version of the model.

Additionally, we wanted to address if AlphaFold Multimer model [37] was capable of predict a correct dimerization for mutations known to produce such outcome (M2, M4). Multimer results are presented on S2 Fig. We observe AlphaFold Multimer predicts dimers for both mutations, providing the same consequence as the predicted for the wild-type protein (i.e., exchange of a single β-strand with its counterpart).

## Molecular dynamics

The overall structure of wild-type antithrombin did not change significantly with simulations at 1000 ns and low (27°C), high (40°C) or extreme temperatures (100°C), with the RCL released, showing a stressed structure. Indeed, the structure was also similar (stressed native) for variants secreted with a relaxed structure (M3 and M5) with minor changes near the

**Table 4. Summary of RMSD metrics of the difference of energy in REU (Rosetta energy units) between the wild-type and each mutant.** Differences between ROSIE 2's predictions and AlphaFold's are presented as RMSD and lDDT.

| Protein | Energy (REU) | RMSD (Å) | lDDT |
|---|---|---|---|
| M1 | 2.27 | 0.987 | 0.8371 |
| M2 | 5.88 | 0.974 | 0.8496 |
| M3 | 1.9 | 1.133 | 0.8272 |

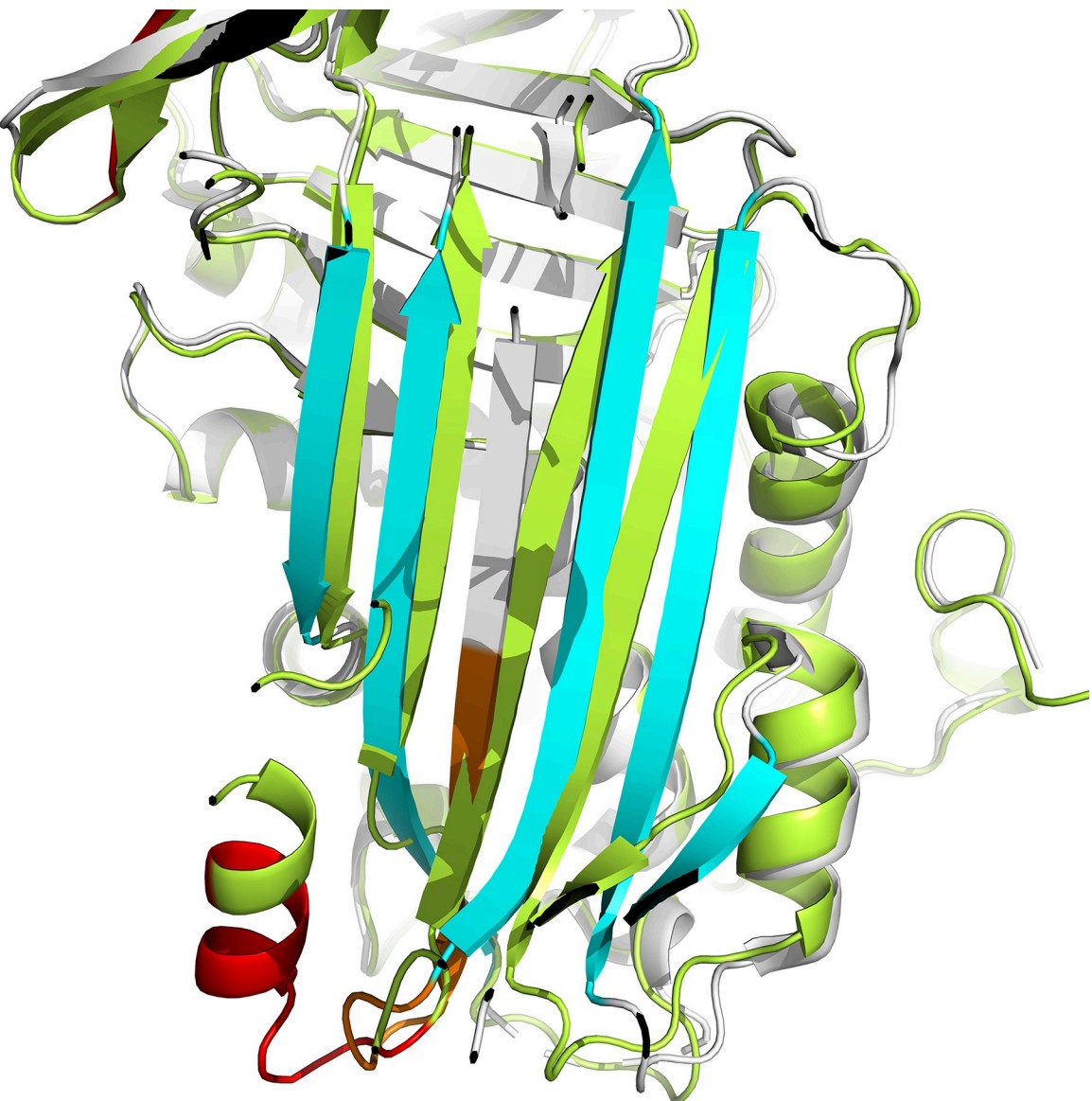

**Fig 5. Cross-section of M5 crystal (4EB1) compared to its AlphaFold prediction.** White: M5 crystal (4EB1). Green: M5 prediction. Cyan: natural serpins β-strands. Orange: 10 amino acid insertion on crystal. Red: 10 amino acid insertion on M5 prediction. 4EB1 shows how M5 insertion leads to an extra β-strand in patients, resembling a latent structure, whilst AlphaFold predicts M5 insertion to be fold as a new α-helix, with no change of the native, wild-type-like structure.

mutations [42]. Metrics are summarized in Tables 3, 4, 8 and 9. PDF reports with all the metrics (RSMD, RSMF and SSE) are included in the supplementary file (S1 File).

We also studied the different proteins using the FEL approach [55] to find the most energetically stable pose within the trajectory and compared its conformation against the initial MD trajectory. These conformations are local minimums. A summary of obtained results is shown in Tables 6 and 7. Details regarding 3D FEL figures are presented in S3 Fig.

## Discussion

The structural flexibility of serpins, required for their efficient inhibitory mechanism, also makes these molecules particularly sensitive to environmental conditions (pH, temperature,

**Table 5. Dihedral angles (Phi/Psi) summary for the wild-type and M5 mutant.** A comparison of crystal structures and AlphaFold predictions.

| Residue | 1AZX | | | 4EB1 | | | M5 AlphaFold | | |
|---|---|---|---|---|---|---|---|---|---|
| | Φ | Ψ | G-Factor | Φ | Ψ | G-Factor | Φ | Ψ | G-Factor |
| Ser204 | -61.12 | 142.8 | -4.287 | 151.14 | 125.93 | -5.91 | -55.45 | 147.54 | -4.503 |
| Glu205 | 73.51 | -10.43 | -10.52 | 158.48 | 169.79 | -6.697 | 64.66 | -9.44 | disallowed |
| Ala206 | -61.86 | -30.56 | -2.958 | 150.61 | 133.67 | -6.326 | -59.39 | -45.66 | -2.772 |
| Ile207 | 116.23 | 139.86 | -4.15 | 122.41 | 131.25 | -3.27 | -69.84 | -37.99 | -2.785 |
| Asn208 | 144.57 | 179.74 | -6.518 | 120.64 | 127.64 | -5.57 | -60.8 | -43.38 | -2.583 |
| Thr211 | 79.4 | 140.8 | -5.42 | -63.58 | 124.65 | -6.222 | 129.23 | 86.68 | -7.127 |
| Val212 | 129.16 | -33.09 | -8.445 | 103.82 | -44.39 | -6.36 | -91.82 | 118.55 | -4.407 |
| Leu213 | -154.8 | 134.71 | -6.027 | 150.11 | 143–98 | -6.076 | 119.73 | 120.64 | -4.606 |
| Val214 | 135.19 | 137.84 | -4.226 | 125.08 | 118.29 | -3.956 | 112.88 | 121.97 | -3.67 |
| Leu215 | 105.47 | 132.19 | -4.114 | -92.02 | 124.16 | -4.712 | -92.25 | 126.78 | -4.665 |

redox conditions) with both physiological and pathological consequences [56–61]. Even minor missense mutations might disturb the stressed native conformation and ease the transition to relaxed hyperstable non-inhibitory conformations with the RCL inserted (mainly latent or polymers) [3]. However, there are other mutations affecting RNA stability or translation leading to quantitative deficiency of serpins and could have pathological consequences. Finally, there are also mutations affecting functional domains that, with minor effects in folding or secretion, cause a qualitative deficiency with inactive variants also involved in different disorders [8]. Hemostatic serpins, particularly antithrombin, are excellent examples of the pathogenic impact of these environmental conditions and types of mutations [3, 9, 62]. Thus, it would be of great interest to have tools able to predict the effect of mutations affecting serpins, particularly those causing transitions to the latent or polymer conformations as they seem to have higher clinical severity [63], probably by a dominant negative effect additional to the loss of function associated to these mutations [64, 65], and by potential additional deleterious consequences if accumulated intracellularly.

In this study, we explored the predictions of AlphaFold, a novel artificial intelligence system conceived to predict structures of new proteins, and molecular dynamics, a computer simulation method for analyzing the physical movements of atoms and molecules on the consequences of five *SERPINC1* mutations causing antithrombin variants, four of them with experimental evidence supporting a conformational change, as well as the effect of high temperatures that caused conformational changes even in wild-type antithrombin [56].

**Table 6. Secondary structure comparison for the wild-type and M5 mutant.** Analysis of crystal structures and AlphaFold predictions.

| Residue | 1AZX | 4EB1 | M5 AlphaFold |
|---|---|---|---|
| *Ser204* | β-strand | β-strand | β-strand |
| *Glu205* | Turn/loop | β-strand | Turn/loop |
| *Ala206* | right-handed α-helix | β-strand | right-handed α-helix |
| *Ile207* | β-strand | β-strand | right-handed α-helix |
| *Asn208* | Turn/loop | β-strand | right-handed α-helix |
| *Thr211* | β-strand | β-strand | β-strand |
| *Val212* | right-handed α-helix | right-handed α-helix | β-strand |
| *Leu213* | β-strand | β-strand | β-strand |
| *Val214* | β-strand | β-strand | β-strand |

**Table 7. Analysis of the hydrogen bond network, a comparative summary.** Comparison of pairs of residues linked by at least one hydrogen bond in each structure. Structure A is the reference on each analysis. Structure B is the corresponding AlphaFold prediction.

| Structure A | Structure B | % shared HBN |
|---|---|---|
| 1AZX-I | Wild-type | 60.00% |
| | M5 | 19.00% |
| | 4EB1-I | 15.00% |
| Wild-type | M1 | 89.00% |
| | M2 | 88.00% |
| | M3 | 89.00% |
| | M4 | 47.00% |
| | M5 | 26.00% |
| 4EB1-I | M5 | 54.00% |

AlphaFold gave an accurate prediction of the wild-type antithrombin structure. Similarly, M1, a recurrent functional mutation that impairs the binding of antithrombin to its cofactor heparin, caused a minor defect in the structure predicted by AlphaFold. However, AlphaFold also maintained the native conformation for all other mutations with experimental evidence supporting a strong conformational consequence, both polymer/dimer formation (M2, and M4), latent transition (M3) or a new relaxed structure formed as a consequence of a relatively large insertion (M5). It has been argued that AlphaFold has not been designed to predict the effect of SNVs [20], but as shown for the cases with 4 to 10 residues insertion, using either reduced or full versions, it only adapts the inserted residues to the structure with minor deviations of the native conformation. Thus, for the in-frame insertion of 4 residues (M4), Alpha-Fold expands a loop to include the new 4 residues, and for the more complex variant M5, a small helix containing the 10 inserted residues was generated.

To check the dimer results for M2 and M4, we ran AlphaFold Multimer [37] (S2 Fig). We found both M2 and M4 do form dimers. Nonetheless, we also find AlphaFold Multimer output

**Table 8. Summary of RMSD metrics between initial and most energetically stable MD states for all studied proteins.** Local minimums were calculated using FEL calculations.

| Protein | Local minima (n) | RMSD (Å) local minimum 1 | RMSD (Å) local minimum 2 | RMSD (Å) local minimum 3 |
|---|---|---|---|---|
| Wild-type | 2 | 1.877 | 1.646 | - |
| M1 | 3 | 1.596 | 1.615 | 1.541 |
| M2 | 3 | 1.820 | 1.832 | 1.718 |
| M3 | 2 | 1.754 | 1.735 | - |
| M4 | 2 | 1.703 | 1.614 | - |
| M5 | 1 | 1.732 | - | - |

**Table 9. Summary of lDDT metrics between initial and state with less energy for all considered proteins.**

| Protein | Local minima (n) | lDDT local minimum 1 | lDDT local minimum 2 | lDDT local minimum 3 |
|---|---|---|---|---|
| Wild-type | 2 | 0.723 | 0.736 | - |
| M1 | 3 | 0.724 | 0.731 | 0.727 |
| M2 | 3 | 0.723 | 0.721 | 0.727 |
| M3 | 2 | 0.717 | 0.734 | - |
| M4 | 2 | 0.723 | 0.746 | - |
| M5 | 1 | 0.717 | - | - |

dimers for wild-type. Indeed, the dimer structure AlphaFold computes form M2 and M4 is the same as the computed for the wild-type sequence, exchanging one β-strand with the A sheet of the other molecule. Moreover, the dimer computed does not correspond to the experimentally determined models (such as PDB: 2ZNH, Fig 1D), in which two β-strands are exchanged between the two counterparts. We include images presenting each scenario below for further clarification on our results regarding AlphaFold Multimer, highlighting in purple the β-strands exchanged.

Regarding ROSIE2 results, we found higher values of REU (more destabilization) for one of the mutations whose mechanism is based on structural disruptions (M2). Still, in contrast, we got lower REU values for M3 (structural disruption) than for M1 (whose mechanism does not change the protein structure). Overall, we find ROSIE2 predictions may help to determine the pathogenicity potential of some mutations, but there seem to be better methods to evaluate missense mutations in serpins altogether.

Molecular dynamics also maintain the native conformation for all variants causing relaxed structures, even creating a new helix for the more complex variant. But the prediction also yielded the native conformation when the experiments were executed under environmental conditions that exacerbated or directly induced conformational changes (transition from native to relaxed) of mutants, and even the wild-type molecule (40˚C and 100˚C).

Thus, these two predictive tools forced the folding of antithrombin to the native stressed conformation even for mutations or conditions that render relaxed structures with experimental evidence.

The explanation for these incorrect predictions may be found in the PDB database. This database contains 334 entities annotated as serpins (with PFAM identifier PF00079), most of them in the native conformation. Indeed, only 36 of those remain when searching for latent. Interestingly, for human antithrombin, the first crystal structure obtained simultaneously by two independent groups contains a dimer of a latent and a native molecule [66–68]. This misbalance of structures with native conformation, and the strong effect of using a specific structure for molecular dynamics studies may lead these predictive tools to force a native structure as the final result of all predictions, even over the own crystal as it occurs for the M5 complex insertion [34]. Therefore, it is necessary to improve predictive tools of folding for serpins, which must consider the conformational sensitivity of these molecules and the transition to hyperstable conformations as the most feasible folding associated with disturbing mutations. In the case of MD simulations, we hypothesize these limitations might be overcome if we could reach longer time scales where such structural rearrangements occur, using specialized and private supercomputers such as ANTON-3 [69], or applying special MD techniques that force sampling of specific events such as accelerated MD or Metadynamics [70–72]. Nonetheless, MD results of all proteins showed no significant differences. If we could run multiple MD with different initial seeds, perhaps we could appreciate changes between MD simulations for each protein. Furthermore, recent implementations of novel machine learning methods coupled with molecular dynamics may improve the research on the proteins' conformational ensemble [11]. As for AlphaFold, an in-depth study of the method and understanding of the code and training process might allow providing protein-targeted predictions for this family of serpin structures with higher accuracy [73].

In this work, we assessed whether AlphaFold could generalize outside of its established range of applicability. It is important to clearly define the capabilities and limitations of Alpha-Fold, given that if the model was able to generate 3D structures of mutated proteins it would be a huge step forward in several areas of Biomedical research. Rare Diseases research is an example, where given the limited number of patients and resources, it is not always feasible to crystallize mutant proteins to assess the pathogenicity of observed variants. In such cases,

having access to a model capable of predicting the structural consequences of mutations would be a great improvement.

Despite other studies analyzed AlphaFold capabilities with point, simulated mutations, this study includes actual variants, found in patients. Furthermore, we include point mutations, like previous studies have done [20], as well as insertions (M4, M5) and a bigger rearrangement (M5, of which we also have access to its crystal, 4EB1 [34]) to assess AlphaFold capability to generalize out of the serpin structures the model has already seen on its training.

Moreover, AlphaFold is partially based on a neural network. Such systems are a "black box model", making it difficult to guess how they make decisions or why they create a certain output. Thus, running it with different inputs (in this case, sequences of different nature, wild-type, point mutations, rearrangements) may give some clues on how the network is internally working. The results of variants M4 and M5 suggest that AlphaFold is creating relatively simple structures as loops or small helices to accommodate the mutated sequence outside of the main structure, so it does not obstruct in the reconstruction of the native templates seen during the model's training.

### Limitations

The scope of this work is a specific protein superfamily with a highly preserved functional structure. Thus, our conclusions may not apply to all proteins, but to those families with conserved structures across its members. Moreover, whilst efforts are being made to predict structural outcomes of missense variants, we also included in this work two INDELs (M4, M5). Current methods to model the structures of these mutations are not fully developed, nor explored, which may have limited out toolkit to work with them herein.

### Conclusions

In summary, we observed that AlphaFold force predictions into the native conformation, even for mutations with experimental evidence of a conformational change. Moreover, these predictions are not capable of transitioning over time, even when simulated with molecular dynamics under extreme conditions leading to mentioned transition, such as extreme heat. Thus, we find necessary to improve predictive strategies for serpins, considering the conformational sensitivity of these molecules. In a broader sense, it is necessary to elaborate on the current structural prediction methods to allow the accurate prediction of mutant proteins overall [14, 16].

### Supporting information

**S1 Fig. Ramachandran plots for structures analyzed.** Mutant structures correspond to AlphaFold prediction for said variant. Plots entitled in caps and italics refer to stated PDB crystal structures.
(TIF)

**S2 Fig. AlphaFold Multimer predictions.** a) M2. b) M4. c) Wild-type. Green and blue: anti-thrombin. Purple: exchanged β-strand.
(TIF)

**S3 Fig. Free Energy Landscape profiles for all studied proteins after analysis of their MD trajectories.** The X and Y axes represent PC1 and PC2 PCA components and the Z axis free energy value. a) Wild-type, b) M1, c) M2, d) M3, e) M4, f) M5.
(TIF)

**S1 File. Reports of MD simulations.** WT, M3 and M5 were simulated at 300K, 313 K, and 373K. These reports contain information about RMSF, RMSD and SSE.
(ZIP)

**S1 Table. Pathogenicity predictions for selected variants.**
(PDF)

**S2 Table. Comparing side-chain conformations of the wild-type and M5 mutant.** An analysis of Chi1 and Chi2 dihedral angles from crystal structures and AlphaFold Predictions.
(DOCX)

**S3 Table. Assessing the quality of the wild-type and M5 mutant structures.** A comparison of G-factors from crystal structures and AlphaFold predictions.
(DOCX)

# Acknowledgments

Pedro Garrido-Rodríguez hold a contract from CIBERER (U765—CB15/00055). Miguel Carmena-Bargueño is a predoctoral student founded by the Plan Propio de Investigación, UCAM. Authors would like to thank to the supercomputing infrastructure of the NLHPC (ECM-02), Powered@NLHPC, by the Plataforma Andaluza de Bioinformática of the University of Málaga, and the Extremadura Research Centre for Advanced Technologies (CETA−CIEMAT) for their support in this study.

# Author Contributions

**Conceptualization:** Pedro Garrido-Rodríguez, Javier Corral.

**Data curation:** Pedro Garrido-Rodríguez, Miguel Carmena-Bargueño, María Eugenia de la Morena-Barrio, Carlos Bravo-Pérez, Rosa Cifuentes-Riquelme.

**Formal analysis:** Pedro Garrido-Rodríguez, Miguel Carmena-Bargueño, Horacio Pérez-Sánchez, Javier Corral.

**Funding acquisition:** Horacio Pérez-Sánchez, Javier Corral.

**Investigation:** Pedro Garrido-Rodríguez, Miguel Carmena-Bargueño, María Eugenia de la Morena-Barrio, Carlos Bravo-Pérez, Belén de la Morena-Barrio, Rosa Cifuentes-Riquelme, Horacio Pérez-Sánchez, Javier Corral.

**Methodology:** Pedro Garrido-Rodríguez, Miguel Carmena-Bargueño.

**Project administration:** Javier Corral.

**Resources:** María Luisa Lozano, Horacio Pérez-Sánchez, Javier Corral.

**Software:** Pedro Garrido-Rodríguez, Miguel Carmena-Bargueño.

**Supervision:** Pedro Garrido-Rodríguez, Horacio Pérez-Sánchez, Javier Corral.

**Validation:** Pedro Garrido-Rodríguez, Miguel Carmena-Bargueño.

**Visualization:** Pedro Garrido-Rodríguez, Miguel Carmena-Bargueño.

**Writing – original draft:** Pedro Garrido-Rodríguez, Javier Corral.

**Writing – review & editing:** Pedro Garrido-Rodríguez, Miguel Carmena-Bargueño, Horacio Pérez-Sánchez, Javier Corral.

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
