## [Decision Letter · Decision Letter 0]

23 Oct 2023

PONE-D-23-28110Analysis of AlphaFold and molecular dynamics structure predictions of mutations in serpinsPLOS ONE

Dear Dr. Corral,

Thank you for submitting your manuscript to PLOS ONE. After careful consideration, we feel that it has merit but does not fully meet PLOS ONE’s publication criteria as it currently stands. Therefore, we invite you to submit a revised version of the manuscript that addresses the points raised during the review process.

Thank you for submitting your manuscript titled "Analysis of AlphaFold and molecular dynamics structure predictions of mutations in serpins" to PlosOne. We appreciate the time and effort you have put into this research.

While the core idea presented in your manuscript is intriguing and holds promise, there are several areas, as noted by our esteemed reviewers, that need further refinement and attention. I highly suggest to address the major revisions highlighted by reviewer 1 and looking forward to consider the revised version of the article in near future.

We look forward to receiving your revised manuscript.

Kind regards,

Soumendranath Bhakat

Academic Editor

PLOS ONE

4. We note that Figure 1, 2, 3, 4, 5 and 6 in your submission contain copyrighted images. All PLOS content is published under the Creative Commons Attribution License (CC BY 4.0), which means that the manuscript, images, and Supporting Information files will be freely available online, and any third party is permitted to access, download, copy, distribute, and use these materials in any way, even commercially, with proper attribution. For more information, see our copyright guidelines: http://journals.plos.org/plosone/s/licenses-and-copyright.

a. You may seek permission from the original copyright holder of Figure 1, 2, 3, 4, 5 and 6 to publish the content specifically under the CC BY 4.0 license. 

Additional Editor Comments:

Dear Authors,

Thank you for submitting your manuscript titled "Analysis of AlphaFold and molecular dynamics structure predictions of mutations in serpins" to PlosOne. We appreciate the time and effort you have put into this research.

Following a rigorous review process, it has been determined that your submission requires a 'Major Revision'. While the core idea presented in your manuscript is intriguing and holds promise, there are several areas, as noted by our esteemed reviewers, that need further refinement and attention.

We believe that with these revisions, your work can be of significant value to our readership. I am genuinely looking forward to reviewing a more polished version of your article in the near future.

Reviewers' comments:

Reviewer's Responses to Questions

**Comments to the Author**

1. Is the manuscript technically sound, and do the data support the conclusions?

Reviewer #1: Yes

Reviewer #2: No

2. Has the statistical analysis been performed appropriately and rigorously? 

Reviewer #1: No

Reviewer #2: No

3. Have the authors made all data underlying the findings in their manuscript fully available?

Reviewer #1: No

Reviewer #2: No

4. Is the manuscript presented in an intelligible fashion and written in standard English?

Reviewer #1: Yes

Reviewer #2: No

5. Review Comments to the Author

Reviewer #1: While the manuscript presents a clear and interesting story for a broad audience, further details, analysis, and clarification is needed across many sections. See attached document for complete comments.

Reviewer #2: In this manuscript, authors have used Alphafold, Rosetta and short molecular dynamics simulations to predict structural properties of 5 mutants/variants of serpins. As it is already known in the field, the authors find that these tools are not powerful enough to predict the effects of mutations on the protein structure. Thus, this study does not provide any new insights. Besides, the authors have not provided sufficient details on the methods and analysis presented in this manuscript. Therefore, I cannot recommend this manuscript for publication. Please find my specific comments below:

1. Authors describe various conformational states of serpins using the terms “native”, “stressed”, “hyperstable”, and “relaxed structure”. However, these terms are not properly defined in the introduction and there is an ambiguity in the usage of these terms in the manuscript. This makes it difficult to understand which structural state is being discussed. Authors must define these terms and be consistent in the usage of these terms throughout the manuscript. Besides, figures showing these structural states is required for the purpose of clarity.

2. In the methods section, it is mentioned that the data was collected from 350 patients, and 135 unique genetic variations were found among 250 samples. What about remaining 100 patients? Further, the authors mention that anti-FXa and anti-FIIa activities measured. Authors must explain anti-FXa and anti-FIIa activities.

3. What was the reference structure used for computing RMSD values given in Table 2?

4. In Table 3, what is the meaning of “RMSD pruned”? What do Structure A and Structure B imply in the context of RMSD values presented in the table?

5. Ultimately, authors conclude that Alphafold and/or short MD simulations alone are not apt for predicting structural effects of mutations which is already known in the field. While the authors acknowledge that large-scale simulations could potentially predict the structural effects of mutations, they haven’t cited a recent study (https://doi.org/10.48550/arXiv.2309.03649) which demonstrates that Alphafold combined with enhanced sampling MD simulations could potentially predict the effect of mutations on the structure and dynamics of a protein.

Further, the authors mention that the structural predictions by Alphafold are heavily biased by the available structures of Serpin in PDB database. However, they do not discuss the studies demonstrating that this limitation can be circumvented by variations in the input parameters provided for the multiple sequence alignment of Alphafold prediction and combining the structural predictions with the enhanced sampling methods (https://doi.org/10.1016/j.sbi.2023.102645, https://doi.org/10.1021/acs.jctc.2c01189 ).

Overall, this study does not provide any new understanding and overlooks existing studies on structural predictions by AlphaFold combined with MD simulations.

6. PLOS authors have the option to publish the peer review history of their article (what does this mean?). If published, this will include your full peer review and any attached files.

Reviewer #1: No

Reviewer #2: **Yes: **Neha Vithani

---

## [Author Response · Author response to Decision Letter 0]

23 Feb 2024

Dear Dr. Chenette,

Thank you for allowing us to answer the questions arising from our draft manuscript entitled “Analysis of AlphaFold and molecular dynamics structure predictions of mutations in serpins”, submitted to PLOS ONE. We appreciate the time and effort that you and the reviewers have invested in providing us with your feedback. We have discussed all your comments and included changes to reflect the most of your suggestions. We have highlighted the changes in the manuscript.

Additionally, here is a point-by-point response to the reviewers’ comments and concerns.

Journal comments

Thank you for your feedback. We have modified the manuscript format to adapt the document to the requirements. We would like you to confirm whether these changes meet the Journal requirements in the new manuscript version.

We apologize for this mistake. We have tried to modify the "Financial Disclosure" on the submission system. Unfortunately, we were not able to do so. We kindly ask the Editors to consider as correct the stated on the Financial Disclosure in the revised manuscript, and on the Funding Information section of the submission portal.

We could not find the mentioned ethics statement outside the Methods section in our manuscripts. We would like to kindly ask the Journal if the novel manuscript version is suitable regarding this point.

 4. We note that Figure 1, 2, 3, 4, 5 and 6 in your submission contain copyrighted images. All PLOS content is published under the Creative Commons Attribution License (CC BY 4.0), which means that the manuscript, images, and Supporting Information files will be freely available online, and any third party is permitted to access, download, copy, distribute, and use these materials in any way, even commercially, with proper attribution. For more information, see our copyright guidelines: http://journals.plos.org/plosone/s/licenses-and-copyright. We require you to either (1) present written permission from the copyright holder to publish these figures specifically under the CC BY 4.0 license, or (2) remove the figures from your submission:

 a. You may seek permission from the original copyright holder of Figure 1, 2, 3, 4, 5 and 6 to publish the content specifically under the CC BY 4.0 license. We recommend that you contact the original copyright holder with the Content Permission Form (http://journals.plos.org/plosone/s/file?id=7c09/content-permission-form.pdf) and the following text: “I request permission for the open-access journal PLOS ONE to publish XXX under the Creative Commons Attribution License (CCAL) CC BY 4.0 (http://creativecommons.org/licenses/by/4.0/). Please be aware that this license allows unrestricted use and distribution, even commercially, by third parties. Please reply and provide explicit written permission to publish XXX under a CC BY license and complete the attached form.” Please upload the completed Content Permission Form or other proof of granted permissions as an ""Other"" file with your submission. In the figure caption of the copyrighted figure, please include the following text: “Reprinted from [ref] under a CC BY license, with permission from [name of publisher], original copyright [original copyright year].”

We find this point may arise from the preprint version of the unreviewed manuscript, available on DOI: https://doi.org/10.1101/2023.01.31.526415. This preprint is authored by us, and we are the copyright owners of these figures. Given the preprint nature of such document, we would like to check if the preprint policy of the Journal is still the detailed in https://journals.plos.org/plosone/s/submission-guidelines#loc-manuscripts-disputing-published-work - Preprints. Nonetheless, following Reviewer 1 advice, we have opted for a major figure redesign, as detailed below herein, in the manuscript and in the submitted figure files. Thus, the figures detailed in this point are no longer present in the manuscript.

Reviewer 1

Review of “Analysis of AlphaFold and molecular dynamics structure predictions of mutations in serpins”

This work seeks to probe whether or not AlphaFold2 (AF2), a structure prediction AI tool, is capable of probing the impact of clinically occurring disease-associated mutations in serpins (serine protease inhibitors). The authors present a biophysical model in which disease-associated mutations cause a conformational change in a model serpin, SERPINC1, and test AF2’s ability to predict these changes. Their results demonstrate a key finding that is relevant to a broad audience: that AF2 remains unable to wholly predict global conformational shifts and changes upon mutation/insertion of residues. I laude the authors’ clarity in presenting a clear-cut story using a biologically interesting model system that has clinical relevance, and their creative use of drawing upon patient data to select mutations. Not only have the authors presented a clear problem and question, they use as many open-source tools as possible to tackle their hypothesis, and even shared their data demonstrating a commitment to Open Science. However, at the current stage the manuscript remains largely devoid of necessary details that must be included prior to publication (as described in major revisions). Furthermore, the figures require significant editing to ensure that a reader can understand the data provided. Lastly, while the authors present a very useful biophysical model to probe the power of AF2, they only provide structural details regarding a single state - it will be critical to include structural depictions of both models to allow the reader to compare and contrast any structural predictions. 

We thank again Reviewer 1 for their suggestions to improve our manuscript. We have updated such sections in the revised manuscript to try and address the ideas the Reviewer presents on this point.

Major Revisions

 1. The authors do not provide any sort of experimental data in the SI or main text about their immunoelectrophoretic experiments, western blots, PAGE results, and beyond, and should provide the data in the manuscript itself, or provide clear citations to find the relevant data. While the use of this method to study the oligomerization and functional behaviors of their clinical mutants is reasonable, the only data provided is a single table with unitless values, and it remains unclear whether these values were obtained using densitometry or other approaches. I would ask that the authors at least include critically relevant blots that convey the functional impact of their top 5 mutants in the manuscript (or supporting information).

We apologize for the unclear way to show the methods used in our manuscript. The main objective of this study was to test the accuracy of AlphaFold following MD predictions for mutations with conformational consequences in antithrombin, an anticoagulant serpin. Thus, we considered that a large description of all biochemical methods used to characterize the aberrant forms of antithrombin identified in patients with the selected SERPINC1 mutations was out of the scope of this manuscript. Moreover, all these methods have been described by our group elsewhere, and all references to these methods are already included in the manuscript and further clarified. Additionally, figures with blots, proteomic or crystallographic data that convey the functional impact of their top 5 mutants in the manuscript generated with carriers of these mutations by our group have been published previously. All these references are included in the manuscript and available for verification. 

 2. I request that the authors greatly expand the methods section, both for their experimental work as well as for their computational studies. At present, the details provided on their immunoelectrophoretic methods, separation and unfolding experiments, and semi-quantitative determination are lacking critical details that prevent any reader from trying to replicate their work. These omissions must be fixed prior to publication.

Again, we apologize for not clearly presenting the references in which experimental data for all variants can be accessed both by reviewers and readers. Following this and the previous points, we have included the references describing these variants in Table 1. We have also included further details on the description of computational methods, adding descriptions and new references for additional details on this matter.

Regarding AlphaFold, we have included additional details: dimerization models were obtained with AlphaFold Multimer. UCSF ChimeraX was used to obtain metrics and figures for AlphaFold results. Pruned RMSD refers to RMSD computed excluding from the process unstructured or flexible structures to avoid potential biases on the RMSD computations. These results were presented using PyMol. Hydrogen bond analysis was performed with ChimeraX as described elsewhere.

For MD, we create a new section of Materials and Methods, to specify how the original structures were preprocessed to run MD. This section is called "Structure processing". Also, we included in the MD methods the tool used to create the water box (System Builder of Maestro Suite).

 3. While their computational model descriptions are sufficiently detailed for replication, it would be useful if the authors were to expand on the structures, they used to start these simulations, a key missing detail. Additionally, the authors should report the implementation of lDDT and RMSD that was used.

We appreciate the referee’s suggestion of the expansion of the explanation about the method to obtain the structures to run the MD. The crystal structures of Wild Type (1AZX-I) and M5(4EB1-I), and the AlphaFold predictions were used to run the MD. As we said in the revised manuscript, all the structures were preprocessed using the Maestro Tool Protein Preparation Wizard to avoid all problems with hydrogens that could crash between them. The proteins were processed to add the charges using the Maestro tool System Builder. 

lDDT:

Regarding the local Distance Difference Test (lDDT) was performed using the lDDT tool of swiss-model (https://swissmodel.expasy.org/lddt/downloads/) , the lDDT score is a measurement of the difference between the local interactions of the two proteins. This tool was used in the CAPS9 (Critical Assessment of protein Structure Prediction 9: https://predictioncenter.org/casp9/index.cgi).

RMSD:

Regarding the Root Mean Square Deviation (RMSD) was calculated using the Maestro tool Simulation Interactions Diagram (SID). This tool takes the first frame as a reference and superimposes all the frames against the reference to evaluate the fluctuation of the protein along the trajectory. The values that are higher than 3 A indicate that the protein is not stable.

The manuscript has been updated to include these explanations.

 4. While the conformational hypothesis the authors present is presented in a logical manner, I would ask that they provide further detail and expand in the introduction on why they think conformation has any basis on the impact of mutations, and that other mechanistic details are not relevant. This detail is needed to demonstrate that the conformational impact of mutations in SERPINC1 are sufficient to predict disease relevance. For instance, if mutations in SERPINC1 instead primarily cause an inability to bind to heparin, resulting in the clinical phenotype, by changing the heparin binding site without any significant conformational changes, prediction of conformational change from the WT state is insufficient to predict clinical relevance. Alternatively, if the primary change is in the likelihood of oligomerization, perhaps a tool like AlphaFold Multimer would be more appropriate (bioRxiv 2021.10.04.463034). The authors current dive into their conformational model would be vastly supported by additional explanation as to why conformation dynamics may play a role here.

We acknowledge the correct and appropriate comment of Reviewer 1. The serpin superfamily has a pretty conserved structure needed for them to function, as their native (stressed) conformation allow them to run a suicidal inhibitory mechanism. These mutations disrupt the native (functional) structure, leading either to latent (relaxed) or dimers / polymers, losing the ability to inhibit via the mentioned suicidal mechanism. In the revised manuscript we have expanded the Introduction to support the functional or conformational consequences of mutations in antithrombin and their clinical effects. 

Additionally, we have run AlphaFold Multimer to test variants causing dimerization of antithrombin. We found both M2 and M4 do form dimers. Nonetheless, we also find AlphaFold Multimer output dimers for wild-type. Indeed, the dimer structure AlphaFold computes form M2 and M4 is the same as the computed for the wild-type sequence, exchanging one β-strand with the A sheet of the other molecule. Moreover, the dimer computed does not correspond to the experimentally determined models (such as PDB: 2ZNH, Fig 1D of revised manuscript), in which two β-strands are exchanged between the two counterparts. We include images presenting each scenario (S2 Fig. of revised manuscript) for further clarification on our results regarding AlphaFold Multimer, highlighting in purple the β-strands exchanged.

 5. Importantly, I would ask that the authors provide *both* structural models of their two conformations (“stressed”, represented by 1AZX in the PDB, and “relaxed”, represented by 4EB1) in their figures and explicitly clarify that the “p.Glu241…” variant was crystalized by the same lab (PDB: 4EB1). At present, it seems like the authors are only presenting figures of the “stressed” conformation and not providing any further figures/structural details on the disease-associated relaxed state, despite it being more predictive of disease-association. It would be important to provide the reader with structural details of both states, and to overlay them alongside the AF2 predictions, to allow for visual comparison. In particular, after examining the 4EB1 and 1AZX structures, it is not immediately obvious what the consequences of the additional strand in the beta-sheet are. Most of the structure is the same, with an RMSD between the chain “I” of both structures ~1.268 Å according to PyMol. I can imagine that there could be significant changes, but making them clearer in the paper is essential.

Again, we would like to thank Reviewer 1 on their comments, as we think their suggestions allowed us to greatly improve our manuscript. Following their advice, we’ve stated tha

---

## [Decision Letter · Decision Letter 1]

26 Mar 2024

PONE-D-23-28110R1Analysis of AlphaFold and molecular dynamics structure predictions of mutations in serpinsPLOS ONE

Dear Dr. Corral,

Thank you for submitting your manuscript to PLOS ONE. I appreciate your patience. We have now completed the reviewing process by an independent reviewer and I am happy to accept the revised manuscript after you address the comments of the reviewer and update the manuscript accordingly. Thanks for understanding and looking forward to receive the updated manuscript.

Best regards,

Soumendranath Bhakat

We look forward to receiving your revised manuscript.

Kind regards,

Soumendranath Bhakat

Academic Editor

PLOS ONE

Reviewers' comments:

Reviewer's Responses to Questions

**Comments to the Author**

1. If the authors have adequately addressed your comments raised in a previous round of review and you feel that this manuscript is now acceptable for publication, you may indicate that here to bypass the “Comments to the Author” section, enter your conflict of interest statement in the “Confidential to Editor” section, and submit your "Accept" recommendation.

Reviewer #3: (No Response)

2. Is the manuscript technically sound, and do the data support the conclusions?

Reviewer #3: Partly

3. Has the statistical analysis been performed appropriately and rigorously? 

Reviewer #3: No

4. Have the authors made all data underlying the findings in their manuscript fully available?

Reviewer #3: Yes

5. Is the manuscript presented in an intelligible fashion and written in standard English?

Reviewer #3: Yes

6. Review Comments to the Author

Reviewer #3: The author of the manuscript has applied some recent tools like AF and Rosetta to predict the structure of wild-type and mutant SERPINS, after addressing some of the reviewer’s comments the manuscript quality has improved. However, still, this manuscript lacks some basic fundamental questions.

1. The authors have used no template and BDF for the structure prediction via AF. But what happens when we use wild type as a template to predict the mutant state? Does it yield a better structure prediction?

2. As we know the basics of any structure prediction is comparing the dihedrals (chi, phi, psi, and omega) of mutant residue predicted with PDB structure. Computing RMSD and RMSFs does not zoom into the effect of mutations. This can be easily done for M5.

3. Is there experimental evidence that validates the REU analysis in Table 4 that all mutations have a destabilizing effect?

4. In Table 2 RMSD was done for which atoms?

7. PLOS authors have the option to publish the peer review history of their article (what does this mean?). If published, this will include your full peer review and any attached files.

Reviewer #3: No

---

## [Author Response · Author response to Decision Letter 1]

10 May 2024

R2: response letter

PONE-D-23-28110R1

Analysis of AlphaFold and molecular dynamics structure predictions of mutations in serpins

Dear Dr. Chenette,

Thank you for allowing us to answer the questions arising from our draft manuscript, "Analysis of AlphaFold and molecular dynamics structure predictions of mutations in serpins, " submitted to PLOS ONE. We appreciate the time and effort that you and the reviewers have invested in providing us with your feedback. We have discussed all your comments and included changes to reflect most of your suggestions. We have highlighted the changes in the manuscript. Here is a point-by-point response to the reviewers’ comments and concerns.

Reviewer 3

1. The authors have used no template and BDF for the structure prediction via AF. But what happens when we use wild type as a template to predict the mutant state? Does it yield a better structure prediction?

First, we thank Reviewer 3 for their comments on how to improve our manuscript.

AlphaFold relies on a custom database designed by DeepMind called Best Fantastic Database (BFD). AlphaFold developers allow users to choose between running the model relying on a full BFD (containing most sequences and structures logged in public databases) or a reduced version of the BFD. Thus, the capabilities of AlphaFold entirely rely on this BFD. In both cases, multiple proteins can affect the final result. 

AlphaFold users cannot manually select a single input for the model to produce a result. Moreover, in that scenario, the output of AlphaFold may not be comparable to the results produced by both complete or reduced AlphaFold for the reason above.

Nonetheless, we have tested results comparing complete and reduced AlphaFold for wild-type and M5 sequences, finding no significant differences between the two results for any of the sequences tested. These results are presented in Table 2 of the manuscript.

2. As we know the basics of any structure prediction is comparing the dihedrals (chi, phi, psi, and omega) of mutant residue predicted with PDB structure. Computing RMSD and RMSFs does not zoom into the effect of mutations. This can be easily done for M5.

We are grateful to the reviewer for their suggestion. We calculated the dihedral angles of the residues near mutation (S204, E205, A206, I207, N208, T211, V212, L213, V214, and L215). When we compared the three previous residues after the mutation, we found differences between the crystal structure of the M5 and the prediction of AlphaFold. The crystal structure has β-sheets before the mutation, whilst the AlphaFold prediction introduces a right-handed α-helix. These results agree with those presented in Tables 5 and 6 and page 16 lines 319-323 (manuscript with tracking). 

We also updated the manuscript with tables (S6 Table and S7 Table) containing information about dihedral angles.

3. Is there experimental evidence that validates the REU analysis in Table 4 that all mutations have a destabilizing effect?

This is an interesting point. As the mutations selected have been studied previously using experimental techniques, we have information on their behavior, mechanisms leading to disease, and other metrics.

For M1, the mechanism is on residue 79 itself. As Arg79 is a critical residue that directly interacts with heparin, the mutation to Cys (losing the positive charge associated with the Arg) severely impairs the interaction between antithrombin and heparin. The differential interaction between the WT molecule and the M1 variant with heparin was proved experimentally by crossed immunoelectrophoresis in the presence of heparin. Figure A shows the differential migration pattern of the molecules bound or not bound to heparin, both in the plasma of a healthy subject and in plasma of a heterozygous carrier of the p.Arg79Cys variant, as well as in recombinant WT and M1 variants. Thus, the pathological mechanism of this mutation is not structure-based, so we would not expect structure disruptions for this mutation. 

Figure A. Electrophoretic assay of WT (Arg79) and M1 (Cys79) antithrombins by their interaction with heparin. Differences of plasma and recombinant antithrombins are shown.

The mechanism involved in antithrombin deficiency is different for mutations M2 and M3. M2 (p.Pro112Ser) leads to deficiency by creating abnormal antithrombin polymers that are mainly retained inside the cells. However, traces of disulphide-linked mutant dimers are detected in plasma, as our group has published elsewhere (J. Corral et al., “Mutations in the shutter region of antithrombin result in the formation of disulfide-linked dimers and severe venous thrombosis,” J. Thromb. Haemost., vol. 2, no. 6, pp. 931–939, 2004, doi: 10.1111/j.1538-7836.2004.00749.x). In this case, the mutation creates a structural disruption, and it is in line with the REU result for M2.

The last missense mutation, M3 (p.Met283Val), leads antithrombin to fold into a latent confirmation, with the RCL inserted as a new beta-strand in sheet A. The variant is secreted but has no anticoagulant activity and low affinity for heparin, as our group demonstrated previously (M. de la Morena-Barrio et al., “High levels of latent antithrombin in plasma from patients with antithrombin deficiency,” Thromb. Haemost., vol. 117, no. 5, pp. 880–888, 2017, doi: 10.1160/TH16-11-0866). There is a structural disruption in this case, but it is not as detected as in the M2 case with the REU metrics.

Thus, we find higher values of REU (more destabilization) for one of the mutations whose mechanism is based on structural disruptions (M2). Still, in contrast, we got lower REU values for M3 (structural disruption) than for M1 (whose mechanism does not change the protein structure). Overall, we find ROSIE 2 predictions may help to determine the pathogenicity potential of some mutations, but there seem to be better methods to evaluate missense mutations in serpins altogether.

We have included these observations in the new version of the manuscript to further clarify ROSIE 2's potential for assessing the pathogenicity of missense variants in serpins.

4. In Table 2 RMSD was done for which atoms?

Table 2 was computed using the heavy atoms of each structure pair, as provided by PyMol RMSD calculations.

We would like to thank Reviewer 3 again for their feedback. Their suggestions have improved the quality of the manuscript.

---

## [Editor Report · Decision Letter 2]

14 May 2024

Analysis of AlphaFold and molecular dynamics structure predictions of mutations in serpins

PONE-D-23-28110R2

Dear Dr. Corral,

Dear Authors,

Thank you for submitting the revised version of PONE-D-23-28110 to PLOS ONE. I am pleased to inform you that your manuscript has been judged scientifically suitable for publication and will be formally accepted for publication once it meets all outstanding technical requirements.

Kind regards,

Soumendranath Bhakat

Academic Editor

PLOS ONE

Additional Editor Comments (optional):

Dear Authors,

Thank you for submitting the revised version of PONE-D-23-28110 to PLOS ONE. After careful consideration, I am pleased to accept the latest version of your manuscript for publication.

Best regards,

Soumendranath Bhakat
---

## [Editor Report · Acceptance letter]

25 Jun 2024

PONE-D-23-28110R2 

PLOS ONE

Dear Dr. Corral, 

I'm pleased to inform you that your manuscript has been deemed suitable for publication in PLOS ONE. Congratulations! Your manuscript is now being handed over to our production team.

Kind regards, 

on behalf of

Dr. Soumendranath Bhakat 

Academic Editor

PLOS ONE